# Assessing Whey Protein Sources, Dispersion Preparation Method and Enrichment of Thermomechanically Stabilized Whey Protein Pectin Complexes for Technical Scale Production

**DOI:** 10.3390/foods10040715

**Published:** 2021-03-27

**Authors:** Jessica M. Filla, Maybritt Stadler, Anisa Heck, Jörg Hinrichs

**Affiliations:** Institute of Food Science and Biotechnology, University of Hohenheim, D-70593 Stuttgart, Germany; Maybritt.stadler@uni-hohenheim.de (M.S.); Anisa.heck@uni-hohenheim.de (A.H.); j.hinrichs@uni-hohenheim.de (J.H.)

**Keywords:** *β*-lactoglobulin, scraped-surface heat exchanger, fat mimetic, process optimization, protein aggregate separation, byproduct, side-stream, dietary fibers

## Abstract

Whey protein pectin complexes can be applied to replace fat in food products, e.g., pudding and yogurt, contributing to creaminess while adding a source of protein and fiber. Production of these complexes is usually conducted on the laboratory scale in small batches. Recently, a process using a scraped-surface heat exchanger (SSHE) has been employed; however, dispersion preparation time, feasibility of using different whey protein sources and enrichment of the complexes for subsequent drying have not been assessed. Preparing whey protein pectin dispersions by solid mixing of pectin and whey protein powders resulted in larger complexes than powders dispersed separately and subsequently mixed after a hydration time. Dispersions without hydration of the mixed dispersions before thermomechanical treatment had the largest particle sizes. The targeted particle size of *d*_90,3_ < 10 µm, an important predictor for creaminess, was obtained for five of the six tested whey protein sources. Dispersions of complexes prepared using whey protein powders had larger particles, with less particle volume in the submicron range, than those prepared using whey protein concentrates. Efficiency of complex enrichment via acid-induced aggregation and subsequent centrifugation was assessed by yield and purity of protein in the pellet and pectin in the supernatant.

## 1. Introduction

The demand for low-fat milk products has increased steadily over the last decade [1]. Unfortunately, these products are often lacking in textural attributes associated with fat, such as smoothness, creaminess and appropriate mouthfeel [2,3]. In the field of protein-based fat replacers, much research has focused on whey protein pectin complexes (WPPC). These complexes are valuable since they combine nutritive properties of whey protein and pectin fibers [4,5] with functional properties, such as organoleptic and structuring abilities, resulting in the perception of creaminess in milk products, e.g., low-fat yogurt [6,7,8]. Heating whey protein pectin dispersions (WPP) above the denaturation temperatures of whey proteins initiates WPPC formation, due to unfolding of *β*-lactoglobulin (*β*-Lg), which exposes the reactive sulfhydryl group. WPPC can be produced with additional shear treatment via thermomechanical treatment of WPP dispersions in a laboratory scale scraped-surface heat exchanger (SSHE) [9,10]. Additional shear treatment gives another possibility to tailor particle size through mechanical force, compared to heating alone.

The particle size of WPPC is an important predictor for creaminess and appropriate mouthfeel when replacing fat [9,11]. For application in milk products, particle sizes of WPPC ranging from 1 to 10 µm evoke the highest perceived creaminess, defining this as the target particle size range [12]. Above 10 µm, single complexes may be perceived as sandy and gritty, while below 1 µm particles cannot appropriately contribute to creaminess and contribute to a watery perception. WPPC formation is initiated via heating at temperatures of 75–90 °C, leading to whey protein denaturation. The whey proteins unfold and reveal charged groups, such as the previously shielded reactive sulfhydryl groups in *β*-lactoglobulin, which then undergo subsequent irreversible aggregation while interacting with pectin [7,12,13,14]. At temperatures of 75–90 °C and pH values above the isoelectric point (pH > 4.5), where strong electrostatic interactions are present, stable complexes are formed at faster rates [15]. With these processing parameters, the properties of WPPC can be tailored in the SSHE [9,16,17]. Additionally, heat-induced complexes formed in the SSHE can be disrupted by shearing, where weaker noncovalent interactions are involved in forming WPPC in a suitable size range [9].

Hydration is the first step that is conducted when whey protein powders are incorporated as functional ingredients. To obtain desired functional properties, powders need to be hydrated prior to heat treatment. Therefore, Protte et al. [18] dissolved protein and pectin powders separately, hydrated these stock solutions overnight, blended them together the next day and stored the mixture again overnight at 8 °C before formation of noncovalent bound coacervates. This approach is very time consuming, limiting the scope of experiments and technical scale-up. 

Previous research on WPPC formation has focused mainly on preparation in lab scale experiments, where WPPC were formed by (mechanical) heat treatment in a water bath with or without magnetic stirring [8,18,19]. Protte et al. [9] upscaled the thermomechanical treatment for WPPC to a laboratory SSHE with a filling volume of around 130 mL, as an intermediate step prior to technical scale application. At technical scale production, SSHE minimum product flows of 130 L h^−1^ require high amounts of WPP. Whey protein powders offer an easy choice compared to using fresh whey from cheese production; many steps are required before fresh whey can be used in complex formation. Nevertheless, recycling of whey, i.e., making use of this side-stream, is desirable, making it necessary to examine different whey protein sources used for WPPC formation. Graf et al. [17] showed that concentrated ideal whey is suitable for WPPC formation, but showed that there are differences in complex assembly and particle size compared to whey protein isolate. Since whey composition depends on various factors, including the type of cheese being processed, milk thermal treatment and whey source, the influence of different whey protein sources should be investigated [20,21]. 

Protte et al. [22] showed a shift in the biopolymer ratio from an initial protein:pectin ratio of 5:1 (*w*/*w*) to a final ratio of 30:1 (*w*/*w*) in the WPPC, in favor of protein components. Results were obtained by means of mass balance after ultracentrifugation and capillary viscosimetry. Of note is that 98% of the pectin was not incorporated into the WPPC formed at 90 °C and pH 6.1, but remained in the surrounding aqueous phase, thereby increasing its viscosity. High viscosities may become a limiting factor for increasing the concentrations of biopolymers and subsequent spray drying of WPPC dispersions for effective distribution of the product. Additionally, unbound pectin in WPPC powders could alter milk product matrices in undesired ways due to the thickening properties of pectin, necessitating the separation of pectin prior to spray drying. To the authors’ knowledge, no WPPC and pectin-rich phase separation approaches in lab or technical scale have been stated and investigated elsewhere. 

The aim of this work was to optimize the production of thermally stabilized WPPC using an SSHE in the lab scale prior to upscaling in the technical scale. It was hypothesized that (i) dispersion preparation time can be shortened by 24 h via solid powder mixing and subsequent hydration, (ii) WPPC can be formed using different whey protein sources resulting in WPPC in the desired particle size range between 1 and 10 µm and (iii) WPPC in the dispersion can be aggregated via acid-induced secondary aggregation and separated using centrifugation. For the investigation of WPPC formation using different whey protein sources, three different whey protein powders and three fresh whey protein concentrates were used.

## 2. Materials and Methods

### 2.1. Materials 

High-methoxyl pectin HMP (CU 201, citrus) was kindly provided by Herbstreith & Fox (Neuenbürg, Germany) and used without further purification. As stated by the manufacturer, the degree of esterification was 71% and the apparent molecular weight was 85 kDa. 

#### 2.1.1. Whey Protein Powders 

Whey protein isolate 895 WPI_nzmp_ was purchased from Fonterra Co-operative Group (Auckland, New Zealand). Whey protein isolate WPI_SaM_ and whey protein concentrate 80% WPC_SaM_ were kindly provided by Sachsenmilch GmbH (Leppersdorf, Germany). Protein contents used for calculations were as follows: 93.0% WPI_nzmp_, 93.1% WPI_SaM_, 79.0% WPC_SaM_ (manufacturer specifications). 

#### 2.1.2. Whey Protein Concentrates

Three whey protein concentrates were investigated in this study. The three production processes are shown in Figure 1. Two types of whey were produced in the Dairy for Research and Training (University of Hohenheim, Stuttgart, Germany): ideal whey W_ideal_ and whey from cut cheese production W_tech_ (see Figure 1A,B). Raw milk was obtained from the Dairy Research Station (Meiereihof, University of Hohenheim, Stuttgart, Germany) and pasteurized (74 °C, 30 s). For whey from directly acidified cut cheese (W_tech_), the fat-to-protein ratio was standardized by mixing skim milk and cream at the required ratios to 0.9 (−). A total of 450 L of prepared milk was cooled (*ϑ* < 10 °C) and enriched with 0.02% (*v/w*) of a 40%-CaCl2 solution (Carl Roth GmbH) and 0.015% (*v/w*) of a 30%-lysozym solution (IP Ingredients GmbH). Afterwards, the milk was acidified to pH 5.7 (−) by addition of a 10% (*w*/*w*) lactic acid solution (AppliChem GmbH, Gaterleben, Germany). After 30 min, the pH was readjusted to 5.7 (−), due to the buffer capacity of milk. The acidified milk was tempered to 35 °C, followed by rennet-induced coagulation initiated by addition of 0.02% (*v/w*) chymosin (200 IMCU mL^−1^; CHY-Max Plus, Christian Hansen, Lübeck, Germany). The coagulum was cut after 10–15 min. Heat treatment of the curd (35 °C; 90 min) under constant stirring (25 rpm; paddle stirrer) was done to promote syneresis. After heating, the curd was transferred to cheese molds for drainage. A total of 350 L of whey was collected and separated from milk fat and cheese fines with a separator. W_tech_ was stored at 8 °C until concentration.

For ideal whey (W_ideal_), 400 L raw milk was skimmed (≤0.1% (*w*/*w*) fat) and preheated to 49.5–50.5 °C in a tank. The skim milk was microfiltrated (MF) using a ceramic membrane with a nominal pore diameter of 0.1 µm and a total membrane area of 1.69 m^2^ (Membralox 7P19-40GP, Pall Exekia, France) on a tangential flow filtration plant (Pall GmbH, Dreieich, Germany). During the filtration, a transmembrane pressure of 0.24 MPa and a filtration temperature of 49.5–50.5 °C were maintained. The microfiltration was performed at a flux of 40 m^3^ h^−1^. Permeate was collected representing ideal whey and immediately concentrated.

Additionally, whey was provided from a local cheese plant W_ind_ (Milchwerke Schwaben eG, Neu-Ulm, Germany). Whey from the production line for semi-hard cheese (Edam) was separated from milk fat and cheese fines and collected over 1 h of production. Afterwards, the whey was shipped to the Dairy for Research and Training (University of Hohenheim, Stuttgart, Germany) pasteurized and concentrated within 4 h after initial production (Figure 1). 

All fresh whey sources were concentrated to protein contents >5% (*w*/*w*) by ultrafiltration (UF). Therefore, whey was preheated to 49.5–50.5 °C in a tank connected to the UF plant. An organic membrane with a nominal pore diameter of 10 kDa and a total membrane area of 13.4 m^2^ (KMS K131, Koch Membrane Systems Inc., Aachen, Germany) on an UF pilot plant (MMS AG Membrane Systems, Urdorf, Switzerland) was used. A transmembrane pressure of 0.14 MPa and a filtration temperature of 49.5–50.5 °C were kept. The UF was performed at a flux of 40 m^3^ h^−1^ and stopped when protein concentrations >5% (*w*/*w*) were reached in the retentate. The protein-rich retentate and the permeate were collected. Protein concentration of the retentate was determined based on the method of Dumas (IDF 185) using a nitrogen analyzer (Dumatherm DT; C. Gerhardt GmbH & Co. KG, Königswinter, Germany) (ISO 14891:2002(E)). The protein content was calculated by multiplying the total nitrogen content by a conversion factor of 6.38. The final protein content of the retentate was adjusted to 5% (*w*/*w*) for all three whey concentrates by diluting it with the permeate from the UF. Retentates and permeates were stored at 8 °C until further use.

### 2.2. Characterization of the Whey Protein Sources

The whey protein sources were characterized using the following methods: the pH was measured at 21 °C according to the standard method for milk and milk products (C 8.2, VDLUFA, 2003). The protein content was evaluated based on the method of Dumas (see Section 2.1.2). The nonprotein nitrogen was evaluated using the standard method for milk and milk products (C30.3, VDLUFA, 1985). Dry matter was determined according to the sea sand method (C35.3, VDLUFA, 2003). 

The quantification of the whey proteins *α*-lactalbumin (*α*-La) and *β*-Lg was performed by reversed phase-HPLC (RP-HPLC) as described by Ostertag et al. [23] using a chromatographic system (Agilent Technologies, Santa Clara, CA, USA) according to IDF 178:2005. Measurements were conducted at 40 °C using a BioResolve RP mAb 450A Polyphenyl column (pore size: 45 nm, particle size: 2.7 μm, 150 mm × 4.6 mm; Waters Corp., Milford, MA, USA). Eluent systems were applied at a flow rate of 0.8 mL min^−1^ with a linear gradient (aqueous eluent: 99% (*v/v*) double distilled water (Aq_DD_: <0.55 µS cm^−1^, Purelab Classic, ELGA LabWater, Celle, Germany) and 1% (*v/v*) Acetonitrile (ACN, purity ≥99.9%, Honeywell International Inc., Charlotte, USA) added with 0.1% (*v/v*) trifluoroacetic acid (TFA, purity ≥99%, Carl Roth GmbH & Co KG, Karlsruhe, Germany); organic eluent: 99% (*v/v*), ACN, 1% (*v/v*) Aq_DD_ added with 0.072% (*v/v*) TFA). The whey protein powders were dissolved in 99% (*w*/*w*) Aq_DD_ to final protein concentrations of 1% (*w*/*w*). The whey protein concentrates with 5% (*w*/*w*) were diluted to 1% (*w*/*w*) by addition of Aq_DD_ (*w*/*w*). The elution profiles were detected with a DAD-detector (Agilent Technologies, Santa Clara,CA, USA) at 210 nm. Standard calibration curves were prepared with protein standards (Sigma-Aldrich Corp., St. Louis, MO, USA) for *α*-La (purity >97%) and *β*-Lg (purity >90%) previously dissolved in distilled water and stored at −28 °C. 

### 2.3. Whey Protein–Pectin Complexes

#### 2.3.1. Preparation of Mixed Whey Protein Pectin Dispersion

For the preparation of mixed whey protein pectin dispersions (as opposed to complexes) using whey protein powders WPI_nzmp_, WPI_SaM_ and WPC_SaM_, stock dispersions of whey protein powders and pectin were prepared with deionized water as described by Protte et al. [18]. The final concentrations were 5% (*w*/*w*) protein and 1% (*w*/*w*) pectin. These concentrations were chosen according to previous studies where WPPC were achieved in the desired particle size range of 1–10 µm [9]. The mass ratio of protein:pectin was kept at 5:1, which was shown to be the optimal ratio in previous experiments [24]. Therefore, WPP dispersions were prepared by mixing protein and pectin stock dispersions prepared with the same volume of 250 mL each with a propeller stirrer at 600 min^−1^, unless stated otherwise. The pH at 21 °C of the six whey protein stock dispersions and the pectin stock dispersion, as well as the WPP dispersions, was determined. The pH of the whey protein dispersions was not adjusted in the present study. Omission of pH adjustment was done to investigate whether the possible use of different whey protein sources without the necessity of pH adjustment is applicable.

The following modified WPP dispersion preparation methods were carried out with WPI_nzmp_ as the whey protein source and are depicted in Figure 2. Mixing (600 min^−1^, 5 min) and hydration (*ϑ* = 8 °C, *t* = 24 h) steps were always performed in the same way. The WPP-A dispersion represents the above described standard preparation. For the WPP-C and WPP-D dispersions, powders were manually solid dispersed, i.e., dry mixed, before mixing with water.

#### 2.3.2. Thermomechanical Treatment

Thermomechanical treatments of WPP dispersions were performed using a lab scale SSHE (technical workshop of the University of Hohenheim, Stuttgart, Germany) as described by Protte et al. [9]. In brief, 130 mL of unheated WPP dispersion was poured into the device. First, shear treatment was started by slowly increasing the representative shear rate ${\dot{\gamma }}_{rep}$ to a constant rate of 675 s^−1^, followed by initiation of the heat treatment. WPP dispersions were heated via a water bath at 88.0–90.5 °C connected to the double jacket of the SSHE for 19.5 min. The heat treatment was applied to achieve protein denaturation of ≥90% [25]. After the heat treatment, the water supply was switched to cooling water (9.8–10.2 °C) for 12 min. The obtained dispersions of WPPC were stored at 8 °C until further analysis. For each whey protein source, three mixed WPP dispersions were heat treated to result in three individual batch replicates of WPPC dispersions.

#### 2.3.3. Enrichment of Whey Protein Pectin Complexes

To obtain secondary aggregation of the WPPC in the dispersion, the pH was adjusted to below the isoelectric point of *β*-lactoglobulin of pH 4.5. Therefore, pH 3.5 and 4.0 (−) were adjusted by addition of 25% HCl (Carl Roth, Karlsruhe, Germany) at 20 °C. For physical separation, WPPC dispersions were centrifuged at 30 °C (Sigma 2-16KL, Sigma Laborzentrifugen GmbH, Osterode am Harz, Germany). Since the process should be optimized for upscaling in the technical scale, a continuous process is necessary. Therefore, lab experiments with parameters applicable for a technical scale decanter (model MD 80-S, Lemitec GmbH, Berlin, Germany) were performed [26]. For the separation of the WPPC from the surrounding pectin-rich phase, the dispersions were centrifuged at 4000× *g* for 10 min, just below the limits of the technical scale decanter. Prior to centrifugation, samples were weighed into centrifuge tubes and preheated in a water bath to 30 °C for 5 min. Centrifugation was performed in duplicate. Supernatant and pellet were carefully separated with a 2 mL glass Pasteur pipette and then weighed. Separated phases were stored at 8 °C until further analysis. 

Figure 3 summarizes the process applied to generate WPPC (see Section 2.3.2) and the enrichment via acid-induced secondary aggregation. 

### 2.4. Characterization of Whey Protein Pectin Complexes

#### 2.4.1. Particle Size Determination

Particle size distributions of WPPC dispersions were determined by static light scattering using an LS 13 320 laser scattering particle size analyzer (Beckman Coulter LS 13 320 fitted with a Universal Liquid Module and control software v6.01, Beckman Coulter Inc., Miami, FL, USA). The calculations are based on the Mie theory allowing particle detection within a range of 0.01–2000 μm. 

The WPPC dispersions were stirred at 350 min^−1^ with a magnetic stirrer for 10 min before measurement. Between 100 and 300 µL of sample was added to the measurement chamber. The pellet, obtained in Section 2.3.3 from centrifugation of samples, was redispersed at a ratio of 1:1 with deionized water using two different methods. Firstly, the pellet was stirred manually with a spatula, and secondly, with a disperser (Polytron PT 2500E, Kinematica AG, Malters, Germany) at 6000 min^−1^ for 60 s. Particle size was evaluated immediately after redispersion and particle size measurements were repeated 24 h after storage at 8 °C. Prior to the measurements on the next day, the dispersions were stirred with a magnetic stirrer at 350 min^−1^ for 10 min. Between 100 and 200 µL of sample was added to the measurement chamber.

An obscuration of between 3 and 7% with maximum Polarization Intensity Differential Scattering (PIDS) of 60% was adhered to for all particle size measurements. All measurements were performed at room temperature (18–20 °C) and each sample was measured in triplicate. 

Particle size distributions are displayed as log-normalized density distributions. The *d*_10,3_, *d*_50,3_ and *d*_90,3_, particle sizes at which 10, 50 and 90% of the particle volume is below, respectively, were calculated for each sample. Additionally, the span was calculated using Equation (1). A real refractive index of 1.48 was determined for WPPC dispersions with different whey protein sources using a method developed by Hayakaw et al. [27]. The real refractive index of the solvent (water) was 1.33. An imaginary refractive index of 0.00 was used for the particles and the solvent [28].
(1)$$span\;=\;\frac{\left(d_{90,3}\;-\;d_{10,3}\right)}{d_{50,3}}$$

#### 2.4.2. Separation Efficiency 

To evaluate the separation efficiency, yield Y and purity P of pectin and protein in the pellet and supernatant were determined according to Equations (2)–(5).
(2)$$Y_{PC}\left(\%\right)\;=\;\frac{m_{PC}}{m_{PC}{}_{0}}\cdot 100$$
(3)$$Y_{P}\left(\%\right)\;=\;\frac{m_{P}}{m_{P}{}_{0}}\cdot 100$$
(4)$$P_{PC}\left(\%\right)\;=\;\frac{m_{PC}}{m_{PC}\;+\;m_{P}}\;$$
(5)$$P_{P}\left(\%\right)\;=\;\frac{m_{P}}{m_{PC}\;+\;m_{P}}\;$$
where $m_{PC}$ and $m_{P}$ are the mass of pectin and protein in g determined in the pellet or supernatant, while $m_{P_{0}}$ and $m_{{PC}_{0}}$ refer to the calculated mass of pectin or protein in g present in the WPPC dispersion before centrifugation. 

Pectin contents of the WPPC dispersions before pH adjustment and centrifugation were evaluated according to Blumenkrantz and Asboe-Hansen [29]. The pectin contents of the supernatant and pellet were also determined. Protein contents of the dispersions (before pH adjustments and centrifugation), pellets and supernatants were determined via the method of Dumas (see Section 2.1.2). 

### 2.5. Statistical Analysis

For *α*-La, *β*-Lg, nonprotein nitrogen content, dry matter, pH, yield and purity, the arithmetic mean values and standard deviations are reported. For particle size parameters *d*_90,3_, *d*_10,3_ and span, mean values and standard error for the three separately prepared WPPC dispersions are reported. Significant differences were analyzed in Minitab (v19.2020.1) using ANOVA or ANCOVA (analysis of covariance using *d*_90,3_ as a covariate) with repeated measures (pH). Significant differences were identified using Tukey’s post hoc test with *α* = 0.05. Significance tested using ANCOVA is emphasized as such in the text.

## 3. Results

### 3.1. Influence of Dispersion Preparation Method on Whey Protein Pectin Complex Formation 

While for WPP dispersion preparation using whey protein concentrate, only powdered pectin needs to be dispersed, preparation of WPP dispersions with whey protein powders, as performed in previous studies, is very time consuming. Therefore, four different methods were carried out to prepare WPP dispersions (methods A–D), differing in the number of hydration and mixing steps. The dispersion preparation method A is the same as that described by Protte et al. [18] and represents the reference method. This method is the most time consuming, with separate stock dispersion preparation for pectin and whey protein, and two hydration periods (each 24 h), namely, before and after mixing the pectin and whey protein dispersions. The WPPC dispersion produced using method A had the smallest particle diameter *d*_90,3_ of 8.3 ± 0.1 µm and narrowest span of 2.6 ± 0.1 (−). 

For method C, where powders were solid dispersed, the WPPC had a particle diameter *d*_90,3_ of 14.1 ± 0.4 µm and span of 4.6 ± 0.2 (−), which were significantly higher than for method A (*d*_90,3_ of 8.3 ± 0.1 µm; span of 2.6 ± 0.1 (−)). In methods B and D, dispersions were not hydrated after mixing and prior to thermomechanical treatment; the resulting WPPC had the largest particle sizes of *d*_90,3_ 28.1 ± 0.1 and 29.2 ± 3.2 µm and largest span values of 8.4 ± 0.2 and 8.9 ± 0.9 (−), respectively. There were no significant differences between particle size parameters of WPPC prepared with separate stock solution hydrated overnight (method B) and solid dispersed powders (method D), both immediately heat treated after mixing. 

In Figure 4, the cumulative particle size distributions of the WPPC prepared using dispersion preparation methods A and B are displayed in an exemplary manner. WPPC produced using all four methods had a similar particle size distribution in the target range between 1 and 10 µm. However, particles were also present in the size distributions of all samples at sizes 10–60 µm. 

### 3.2. Impact of Protein Source on Whey Protein Pectin Complexes

To obtain a more detailed understanding of how the source of whey protein influences the size of particles in WPPC dispersions, the basic production conditions (manufacturers’ specifications) and composition of the three powders WPI_nzmp_, WPI_SaM_ and WPC_SaM_, as well as the whey protein concentrates W_tech_, W_ideal_ and W_ind_, are shown in Table 1. 

The pH depends on prior treatment of the whey protein sources and is significantly different between all investigated samples, except for WPI_SaM_ and WPC_SaM_ (Table 1), representing a wide variety of possible pH values. The whey protein powder dispersions had neutral pH values of 6.87–6.94 (−), since they are all commonly manufactured from sweet whey (Table 1). The pH of whey protein concentrates depends on previous cheese manufacturing parameters, where W_tech_ had the lowest pH of 5.54 (−), since it was obtained as byproduct of directly acidified cut cheese. 

To apply WPPC as fat replacers in milk products, certain conditions must be met. An important predictor for creaminess, apart from viscosity, is the particle size with a desired range of 1–10 µm for WPPC in fermented milk products [9,11,12]. WPPC of all whey protein sources, except for WPI_SaM_ (15.4 ± 1.1 µm), had a particle diameter *d*_90,3_ below 10 µm, fulfilling the particle size requirement (Table 2, Figure 5). WPI_SaM_ results in significantly larger WPPC compared to WPC_SaM_ with *d*_90,3_ of 5.9 ± 0.7 µm, although it was manufactured via the same process and manufacturer. Particle sizes *d*_90,3_ of WPPC prepared with whey protein concentrates tended to be smaller, with 1.9 ± 0.1 µm for W_ind_ and 5.7 ± 1.0 µm for W_ideal_ (Table 2). The particle size *d*_10,3_ is significantly smaller for WPPC made using whey protein concentrates W_ideal_, W_tech_ and W_ind_, as well as WPC_SaM_, with a maximum diameter *d*_10,3_ of 0.7 ± 0.2 µm (Table 2). 

### 3.3. Enrichment of Whey Protein Pectin Complex Aggregates

For further processing of WPPC dispersions, enrichment is necessary since low concentration of WPPC present in the dispersion transfers high amounts of water into the aimed formulation or results in high energy consumption during subsequent drying. Therefore, the enrichment of WPPC by acid-induced secondary aggregation was studied. Suitable pH and temperature for subsequent centrifugation were determined in preliminary experiments. Results showed that a pH below the isoelectric point of *β*-Lg, around 4.5, is necessary to induce aggregation of WPPC. The yield and purity of the protein in the pellet and pectin in the supernatant were determined based on the assumption that WPPC aggregates are formed and separated on the bottom, while unbound pectin and whey protein remains in the supernatant. The results are depicted in Table 3. 

All WPPC, except for W_ind_ at pH 4.0 (−), could be successfully separated into a white gel-like pellet and a turbid to clear supernatant. For WPPC with WPI_nzmp_ and W_tech_, poor phase separation was observed at pH 4.0 (−) resulting in a liquid pellet, while at pH 3.5 (−) a yogurt-like pellet was obtained. WPI_SaM_ and W_tech_ had low yields of protein in the pellet at both pH values, with yields of 31.2 to 55.0%. W_ind_ had the lowest yield at pH 4.0 (−) with 11.0%. WPI_nzmp_, WPC_SaM_, W_ideal_ and W_ind_ at pH 3.5 (−) showed high yields from 63.3 to 72.6%. The highest yield was obtained for WPI_nzmp_ at pH 4.0 (−) with 90.8%. Purity was high in all samples, with the lowest values of 78.6% for WPI_nzmp_ at pH 3.5 (−).

Yield of pectin in the supernatant was low for WPI_nzmp_, WPC_SaM_, W_ideal_ and W_ind_ at pH 3.5 (−), while WPC_SaM_ also showed low yields at pH 4.0 (−) ranging between 13.8 and 46.9%. Purity of pectin in the supernatant was low for most samples, ranging up to 26.6%, except for WPI_nzmp_ and W_ideal_ at pH 4.0 (−) with 59.1 and 53.4%. 

### 3.4. Redispersibility of the Enriched Whey Protein Pectin Complex Aggregates

For the obtained pellets, an appropriate redispersibility is required for further application and processing, such as spray drying. Figure 6 shows the particle size distributions of the redispersed pellets of WPPC prepared using WPI_nzmp_ separated at pH 3.5 (−), dispersed using different methods, compared to the particle size distribution before centrifugation. Results show that the particle size distribution shifted from the initial particle size *d*_90,3_ of the WPPC of 8.3 ± 0.1 µm towards larger particles after redispersion, with particle sizes *d*_90,3_ of 77.5 ± 2.6 µm for manual dispersion and 9.0 ± 0.1 µm for shear treatment. 

## 4. Discussion

### 4.1. Influence of Dispersion Preparation Method on Whey Protein Pectin Complex Formation 

The aim of comparing the preparation methods was to optimize the dispersion preparation of whey protein powders, i.e., reduce the required time, while obtaining particle sizes in the desired range of 1–10 µm. Reference method A resulted in particle sizes within the desired range for WPPC with good fat-replacing abilities (Table 2) [12]. Particle sizes between 1–10 µm and narrow span, both set as requirements for WPPC, were not obtained for methods B–D. This is due to the presence of larger particles in the range between 10 and 60 µm for the WPPC formed using the three alternative dispersion preparation methods B–D (Figure 4). Therefore, hydration time could not be shortened to 24 h. In comparison to method A, the omission of the second hydration time for the mixed WPP dispersion resulted in the largest complexes being formed (methods B and D). Furthermore, solid dispersion of whey powder with pectin requiring longer hydration times was associated with the formation of larger complexes (methods C and D). Often solubility can be improved by the blending of two different powders, thus increasing surface area of the poorer soluble compound [30]. However, results indicate that blending of powders does not shorten hydration time of pectin to 24 h. Observations can be explained by pectin, since it has the highest hydration at acidic pH of 4.6 (−), whereas mixtures had pH values of 6.43–6.45 (−) [31]. Therefore, separate hydration should be preferred, since pectin solutions have a lower pH. Additionally, a hydration step prior to heat treatment is important to avoid particles larger than 10 µm. Prior to heat treatment, WPP coacervates are formed that regulate the particle size distribution of the WPPC by stabilizing potential complexes via electrostatic and hydrophobic interactions [18,19]. 

In conclusion, method C can be applied for WPP dispersion preparation in further experiments in the technical scale with the modification of a separate powder dispersion preparation for good pectin solubility without separate hydration. Thereby, dispersion preparation time can be shortened by 24 h to a total of 48 h. 

### 4.2. Impact of Protein Source on Whey Protein Pectin Complexes

A desired particle size range of 1–10 µm for WPPC destined for use in fermented milk products has been defined [9,11,12]. Nevertheless, the target particle size range can shift depending on food matrix [32,33]. WPPC prepared from all whey protein sources except WPI_SaM_ (15.4 ± 1.1 µm) had a particle diameter *d*_90,3_ below 10 µm, fulfilling the particle size requirement for application in fermented milk products (Table 2, Figure 5). The significantly larger WPPC of WPI_SaM_ compared to WPC_SaM_ can be explained by the smaller *β*-Lg to *α*-La ratio of 2:1 for WPI_SaM_ or high *α*-La of 11.6 ± 1.2% (*w*/*w*) resulting in larger particles compared to WPC_SaM_ (Table 1). Havea et al. [34] demonstrated that the free thiols of heat-treated *β*-Lg or bovine serum albumin (BSA) catalyze the formation of a range of monomers, dimers and higher polymers of *α*-La. Although particles are larger, Engelen et al. [32] and Hahn et al. [33] showed that particle sizes of up to 20 and 40 µm can occur in pudding and fresh cheese, respectively, without negatively affecting sensory properties, such as graininess. 

Particle sizes *d*_90,3_ of WPPC prepared with whey protein concentrates tended to be smaller, in contrast to Graf et al. [17], where larger particle sizes of WPPC using W_ideal_ were found than for WPI_nzmp_, using the same WPI_nzmp_ product and preparation method of W_ideal_. However, in the previous study, both whey protein sources were adjusted to a pH of 6.1 (−) before use. WPPC dispersions with whey protein concentrates had significantly lower pH values after heat treatment that ranged between 5.26 and 6.39 (−), compared to whey protein powders, ranging between 6.44 and 6.45 (−) (Table 2). Additionally, the particle size *d*_10,3_ was significantly smaller for WPPC prepared using whey protein concentrates W_ideal_, W_tech_ and W_ind_, as well as WPC_SaM_, with a maximum diameter *d*_10,3_ of 0.7 ± 0.2 µm (Table 2). Such particles below 1 µm are assigned to single whey protein aggregates that are not incorporated into the WPPC, mainly made up of *β*-Lg, and have been observed in many studies [12,35]. The bimodal particle size distributions, especially pronounced for WPPC made by using whey protein concentrates (Figure 5), are attributed to these single whey protein aggregates. Therefore, use in beverages such as drinkable yogurt is a more suitable application [7]. Increased thermal stability, due to pectin addition as observed by Protte et al. [18] between pH 5.0 to 6.1 (−) resulting in unfinished denaturation and therefore smaller aggregates, can be excluded since all dispersions had a degree of denaturation >95% (data not shown). 

The results show possible application of all protein sources except WPI_SaM_ at the applied thermomechanical conditions (90 °C, 19.5 min, 675 s^−1^) without pH adjustment in different milk products.

### 4.3. Enrichment of Whey Protein Pectin Complex Aggregates

In general, all whey protein sources are suitable for application in different milk products without pH adjustment. For the enrichment of the WPPC via centrifugation, different criteria must be met to ensure good enrichment. Since high yields of WPPC in the pellet are desired, representing the WPPC, protein yields >70% in the pellet and low pectin yields <40% in the supernatant are set as the first criteria. The purity should be around 80% for protein in the pellet, representing the ratio of whey protein to pectin (5:1) in the WPP dispersions, assuming no shift in the ratio of whey protein to pectin. Analogously, a purity of around 20% pectin in the supernatant is expected for suitable samples. 

WPI_nzmp_, W_tech_ and W_ind_ at pH 4.0 (−) showed poor or no visual separation. Results were confirmed by protein yields and purities in the pellet together with WPI_SaM_ and W_tech_ at pH 3.5 (−) yielding <70%. Therefore, set criteria were not met, excluding samples from further consideration. Torres et al. [36] centrifuged whey protein complexes at 3000× *g* for 10 min, thereby separating particles in the range between 1 and100 µm, resulting in a minimal necessary aggregation size of 1 µm. Additionally, those whey protein dispersions were prepared without pectin, which increases the viscosity considerably, affecting separation efficiency of small aggregates. Therefore, significantly lower yields at pH 4.0 (−) might be attributed to insufficiently large aggregates formed before the separation at the conditions in this work (4000× *g*, 10 min, 30 °C) or to high amounts of single denatured whey protein particles after the thermomechanical treatment in the SSHE. An ANCOVA (R^2^_adj_ = 0.116) was conducted to investigate differences between the purity of whey protein at the two tested pH levels. The purity of whey protein for pH 4.0 was significantly higher than for pH 3.5 (*p* = 0.019), where particle size was not a significant covariate (*p* = 0.490). Therefore, particle size *d*_90,3_ does not significantly affect purity for the tested conditions. The low R^2^_adj_ is attributed to non-normally distributed residuals in terms of the collected order of the data. WPC_SaM_, W_ideal_ and W_ind_ at pH 3.5 (−) met the first criterion with protein yields of >70% and protein purity in the pellet of around 80%, ranging between 84.5 and 87.2%. 

The last criteria that must be met are the yield and purity levels of pectin in the supernatant, with desired values of <40 and approximately 20% for yield and purity, respectively. These requirements were only met by WPC_SaM_ and W_ideal_ at pH 3.5 (−) with yields of 40.4 and 35.3% and purities of 23.1 and 19.3%, respectively. Nevertheless, WPI_nzmp_ and W_ind_ at pH 3.5 (−) might be suitable as well, since they showed good overall yields and purities. When comparing the supernatant of the WPPC dispersions, the yield and purity of pectin were significantly higher for separation at pH 3.5 (−) as compared to 4.0 (yield: R^2^_adj_ = 0.974; *p* = 0.001; purity: R^2^_adj_= 0.985; *p* = 0.001; ANOVA with repeated measures on pH). Proteins that form too small aggregates or single denatured whey proteins cannot be separated at the applied conditions. Additionally, due to the protein:pectin ratio of 5:1, remaining protein carries greater weight in the determination of purity.

### 4.4. Redispersibility of the Enriched Whey Protein Pectin Complex Aggregates

For the obtained pellets, appropriate redispersibility is required for further processing. Results showed that a mechanical treatment, i.e., shear, needs to be applied to disrupt noncovalent bonds, resulting in smaller aggregates [1]. After redispersion with shear treatment (1 min, 6000 rpm), the particle size was still be larger than the initial particle size, making application of higher shear rates or times necessary. Since the particle size did not significantly change after one day of storage, shear treatment for redispersion might be a further convenient step to tailor WPPC particle size. Nevertheless, pH needs to be neutralized before drying for a neutral pH of the product, which might also improve redispersibility by increasing solubility of the formed aggregates below the isoelectric point. 

Further, neutralization of pH, stability of these complexes and the minimal particle sizes achievable via shear treatment need to be studied.

## 5. Conclusions

This study showed that different whey protein sources are suited for the production of WPPC by applying a thermomechanical treatment with a lab SSHE. The composition of the six different protein sources depended on previous treatment. The pH was determined by previous cheese manufacturing and linked to the acidification process. Dispersion preparation experiments showed that it is challenging to shorten the preparation method that consists of multiple steps. Hydration prior to heat treatment was crucial to avoid formation of particles larger than 10 µm. Additionally, solid dispersion was not a suitable approach, since more pronounced formation of particles larger than 10 µm occurred, resulting from poor solubility of pectin at the higher pH of the powder blend (approximately pH 6.45 (−) vs. an optimum pH of 4.6 (−) for pectin). Five whey protein sources were suitable for application in producing WPPC with particle sizes in the target range of 1–10 µm to provide, e.g., creaminess in low-fat fermented milk products. WPPC could be successfully separated into a white gel-like pellet and a turbid to clear supernatant. Yield and purity of the pellet and supernatant were significantly influenced by pH, but not by the initial particle size of the WPPC. The redispersion of the pellet for subsequent processing, e.g., spray drying, is possible by applying mechanical treatment like shear, disrupting noncovalent bonds of the WPPC aggregates. 

Before application in the technical scale, yield and purity need to be adjusted in the best way possible, to provide a suitable starting point for upscaling at the decanter. Therefore, phase separation needs to be further investigated at pH values ≤3.5 (−) regarding WPPC formation efficiency in the context of purity, yield, particle size of the aggregates and influence of pectin on viscosity, possibly resulting in larger aggregates and sharper separation. Additionally, redispersed WPPC should be studied in order to determine the obtainable minimal particle size (distribution) after neutralization. In parallel experiments on a technical scale SSHE, expanded setting options like representative shear rate and temperature are to be conducted to validate the results obtained in lab scale. Furthermore, voluminosity of the WPPC will be assessed as an additional parameter to characterize the suitability as a fat replacer.

## Figures and Tables

**Figure 1 foods-10-00715-f001:**
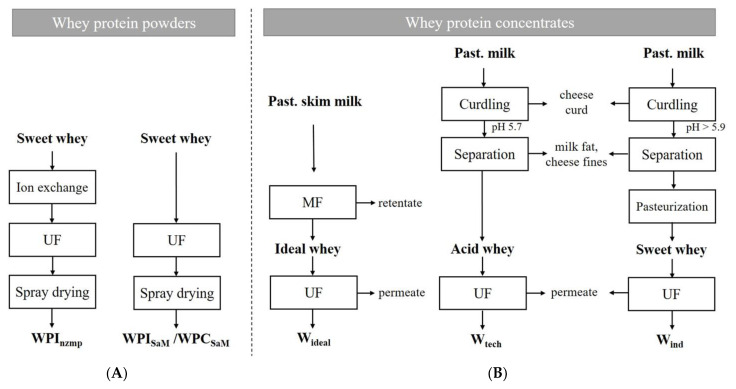
Flowchart of the five processes used for the generation of whey protein powders (**A**) WPI_nzmp_, WPI_SaM_ and WPC_SaM_ and whey protein concentrates (**B**) W_ideal_, W_tech_ and W_ind_ (UF: ultrafiltration; MF: microfiltration).

**Figure 2 foods-10-00715-f002:**
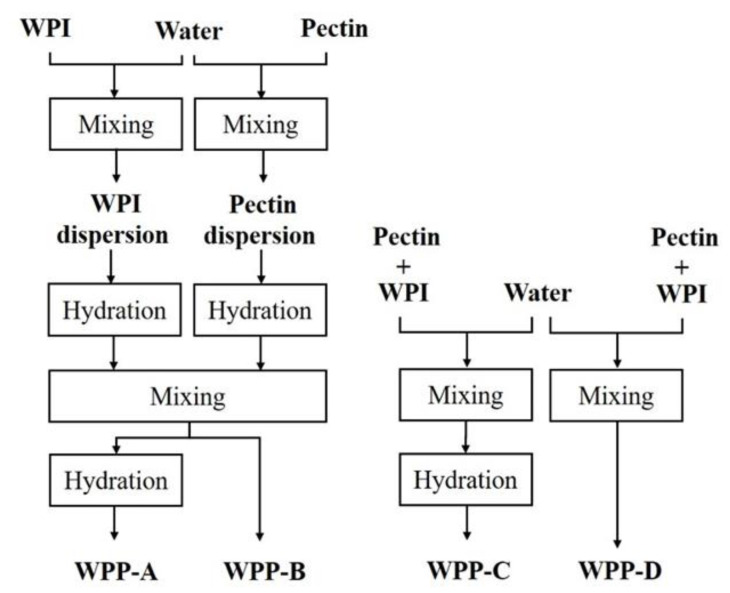
Flow chart of the four methods (A–D) of whey protein pectin (WPP) dispersion preparation with different combinations of hydration (*ϑ* = 8 °C, *t* = 24 h) and mixing steps (A: standard dispersion preparation).

**Figure 3 foods-10-00715-f003:**
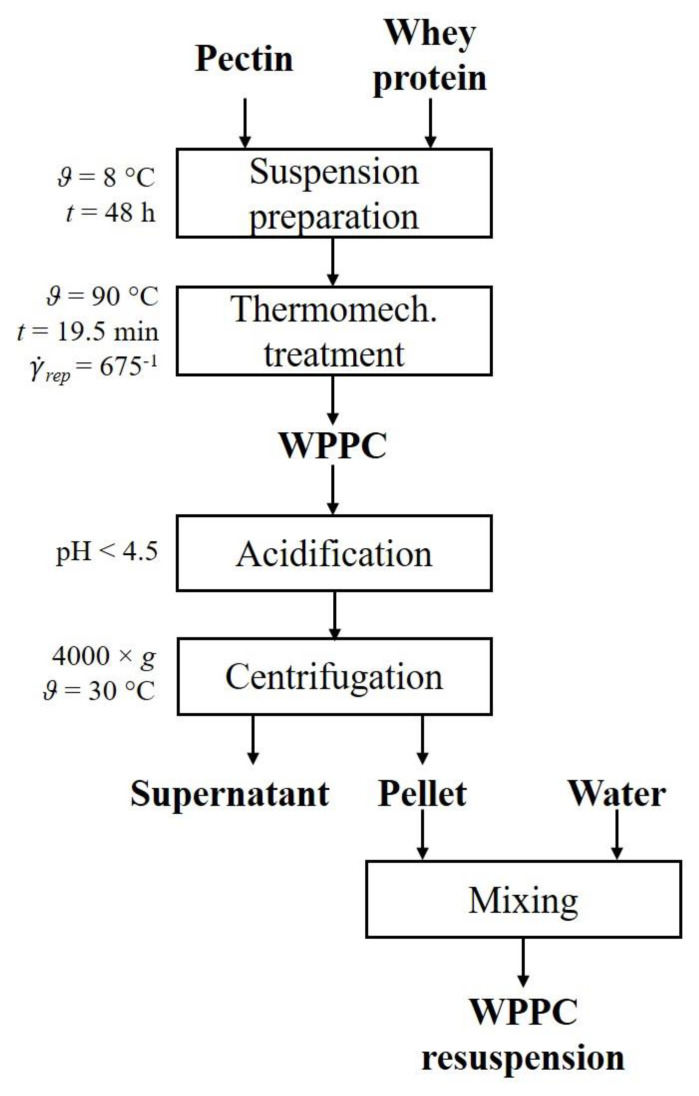
Flow chart of the whey protein pectin complex (WPPC) formation using a scraped-surface heat exchanger (SSHE), their separation and redispersion of the pellet.

**Figure 4 foods-10-00715-f004:**
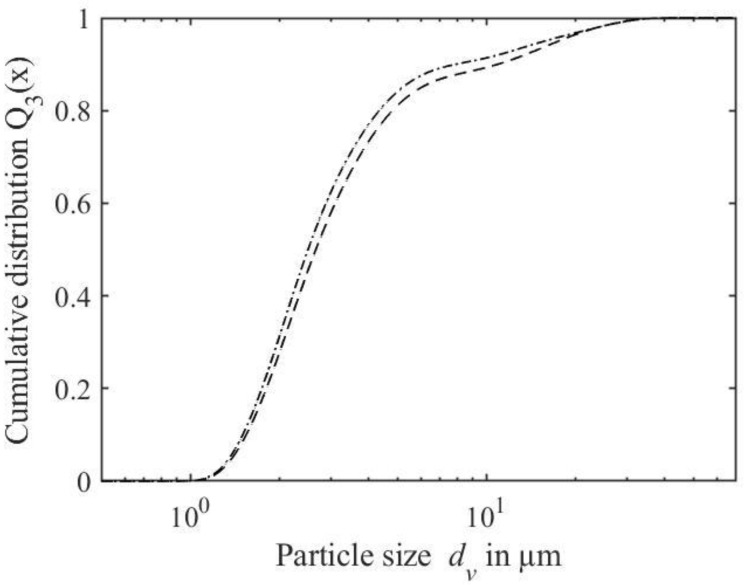
Particle size distributions of whey protein pectin complexes prepared using WPI_nzmp_ and pectin as separate stock solutions (5% protein, 1% pectin) with hydration (method A, solid line) and without hydration (method B, dotted line) after mixing.

**Figure 5 foods-10-00715-f005:**
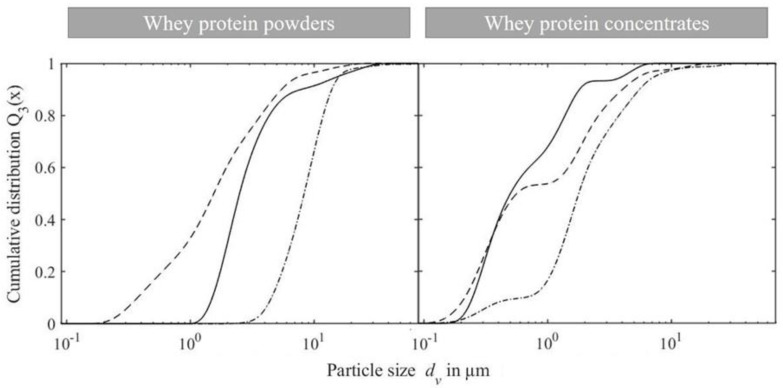
Particle size distributions of whey protein pectin complex dispersions prepared with whey protein powders WPI_nzmp_ (solid line), WPI_SaM_ (dash dot line), WPC_SaM_ (dashed line) and whey protein concentrates W_tech_ (dashed line), W_ideal_ (dash dot line) and W_ind_ (solid line) as whey protein sources mixed with pectin, followed by thermomechanical heat treatment (90 °C, 675 s^−1^, 19.5 min).

**Figure 6 foods-10-00715-f006:**
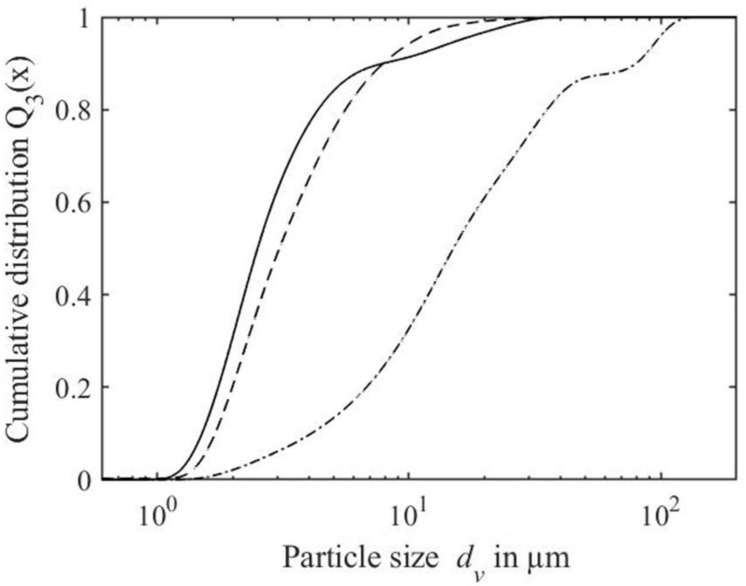
Particle size distribution of whey protein pectin complex dispersions prepared using WPI_nzmp_ before centrifugation (solid line) and after redispersion of the pellet (obtained after centrifugation (4000 × *g*, 10 min, 30 °C, pH 3.5(−)) dispersed via shear treatment of 6000 min^-1^ for 1 min (dashed line) and manual dispersion with a spoon (dash dot line).

**Table 1 foods-10-00715-t001:** Raw material characteristics and composition of dispersions of whey protein powders and whey protein concentrates.

	Raw Material Characteristics			Whey Protein Composition		
Protein Source	pH_21 °C,disp_ ^3^ -	Protein % (*w*/*w*)	TM % (*w*/*w*)	*α*-La ^1^ % (*w*/*w*)	*β*-Lg ^1^ % (*w*/*w*)	NPN ^1^ % (*w*/*w*)
WPI_nzmp_ ^2^	6.94 ^A^	93.0	92.9	0.93 ^A,B^	2.78 ^A^	0.21 ^B^
WPI_SaM_ ^2^	6.87 ^B^	89.0	94.8	1.16 ^A^	2.28 ^B^	0.20 ^B^
WPC_SaM_ ^2^	6.89 ^B^	78.7	95.7	0.58 ^C^	2.28 ^B^	0.21 ^B^
W_ideal_ ^2^	6.70 ^C^	4.2	9.9	0.88 ^A,B^	2.48 ^A,B^	0.37 ^A,B^
W_tech_ ^2^	5.54 ^E^	6.0	8.8	0.63 ^BC^	2.30 ^B^	0.23 ^B^
W_ind_ ^2^	6.23 ^D^	8.1	4.3	0.51 ^C^	1.69 ^C^	0.44 ^A^

^1^*α*-La: *α*-lactalbumin, *β*-Lg: *β*-lactoglobulin, NPN: nonprotein nitrogen in 5% (*w*/*w*) whey protein dispersion. ^2^ WPI_nzmp_: whey protein isolate nzmp, WPI_SaM_: whey protein isolate Sachsenmilch, WPC_SaM_: whey protein concentrate Sachsenmilch, W_ideal_: whey concentrate ideal whey, W_tech_: whey concentrate technical scale cheese production, W_ind_: whey concentrate provided by Schwabenmilch. ^3^ of 5% (*w*/*w*) whey protein dispersion. ^A–E^ Values with different superscript uppercase letters in one column are significantly different (*p* <0.05).

**Table 2 foods-10-00715-t002:** Particle size parameters and pH at 21 °C of dispersions of whey protein pectin complexes prepared using different protein sources.

Protein Source	*d*_90,3_ µm	*d*_10,3_ µm	Span -	pH_disp, 21 °C_ -
WPI_nzmp_	8.3 ± 0.1 ^B^	1.6 ± 0.1 ^B^	2.6 ± 0.1 ^C^	6.45 ^A^
WPI_SaM_	15.4 ± 1.1 ^A^	4.7 ± 0.1 ^A^	1.2 ± 0.1 ^D^	6.44 ^A^
WPC_SaM_	5.9 ± 0.7 ^B,C^	0.4 ± 0.1 ^C,D^	3.3 ± 0.1 ^B^	6.44 ^A^
W_ideal_	5.7 ± 1.0 ^B,C^	0.7 ± 0.2 ^C^	2.5 ± 0.5 ^C^	6.39 ^B^
W_tech_	4.2 ± 0.1 ^C,D^	0.2 ± 0.1 ^D^	6.7 ± 0.1 ^A^	5.26 ^D^
W_ind_	1.9 ± 0.1 ^D^	0.3 ± 0.1 ^D^	3.1 ± 0.1 ^B,C^	5.79 ^C^

Values with different superscript uppercase letters in one column are significantly different (*p* < 0.05).

**Table 3 foods-10-00715-t003:** Yield and purity of the pectin in the supernatant and protein in the pellet separated from dispersions of whey protein pectin complexes prepared using six different whey protein sources.

		Pellet (Protein)		Supernatant (Pectin)	
Whey Protein Source WPPC	pH -	Yield ^1^ %	Purity ^2^ %	Yield ^1^ %	Purity ^2^ %
WPI_nzmp_	3.5	63.3 ^C^	78.6 ^G^	13.8 ^F^	6.9 ^G^
	4	90.8 ^A^	92.79 ^BCDE^	65.4 ^C^	59.1 ^A^
WPI_SaM_	3.5	55.0 ^D^	94.2 ^ABCD^	83.1 ^B^	26.8 ^C^
	4	50.6 ^D,E^	87.9 ^D,E,F^	59.0 ^C,D^	16.7 ^F^
WPC_SaM_	3.5	72.7 ^B^	85.9 ^E,F^	40.4 ^E^	23.1 ^C,D^
	4	68.4 ^B,C^	87.2 ^D,E,F^	39.6 ^E^	17.3 ^E,F^
W_ideal_	3.5	70.4 ^B^	84.5 ^F,G^	35.3 ^E^	19.3 ^D,E,F^
	4	74.7 ^B^	100.5 ^A^	102.1 ^A^	53.4 ^B^
W_tech_	3.5	47.5 ^E^	98.1 ^A,B,C^	95.3 ^A,B^	26.6 ^C^
	4	31.2 ^F^	99.6 ^A,B^	99.2 ^A,B^	22.2 ^C,D,E^
W_ind_	3.5	72.6 ^B^	87.2 ^D,E,F^	46.9 ^D,E^	25.5 ^C^
	4	11.0 ^G^	92.1 ^C,D,E^	95.0 ^A,B^	17.6 ^E,F^

^A–G^ Values with different superscript uppercase letters in one column are significantly different, ANOVA (*p* < 0.05). ^1^ Yield is defined as the percentage of the actual yield determined in the corresponding phase divided by the theoretical yield mathematically determined based on initial concentration. ^2^ Purity is defined as percentage of the actually desired compound in the corresponding phase divided by the total amount of compounds in the phase.

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
