# Peer review of "Assessing Whey Protein Sources, Dispersion Preparation Method and Enrichment of Thermomechanically Stabilized Whey Protein Pectin Complexes for Technical Scale Production"

_foods, 2021, doi:10.3390/foods10040715_

Round 1

Reviewer 1 Report

The experimental design for this study was well planned and the manuscript has been well drafted. There are, however,  a couple of revisions that the authors need to consider. I have highlighted some comments for the authors below:-

Introduction

This section has been clearly written and the scientific gap very well-identified.

Materials and methods

L 129, L226, 266, why was Wtech stored at 8 °C and not any other temperature

L155, Could the authors please explain how the protein content was adjusted to 5 g 100 g-1

for 155 all three whey concentrates

L183 How were the samples were diluted to a final protein concentration of 1 % (w/w)? what solvent was used?

L187 Why -28 °C and not -80 °C or -18 which is the typical temperature for most freezers?

L270 The authors report using a temperature of 18-20 degrees but earlier on they mention 21 degrees. Why were there variations and was there any impact of temperature on the experiments?

L450 The authors mention that the pectin was poorly water-soluble. Could they please quantify this? what was the % solubility of the pectin? And what was the temperature of water used in this case?

L451, Please consider revising the sentence “This indicating that blending of powders does not improve the solubility to the required extant” to make it clearer.

L491 “Therefore, use in beverages like drinking yogurt is a possible suitable” Please revise

L568 Why do the authors propose investigating phase separation at pH ≤ 3.5 (-)?

Author Response

Detailed response to comments of reviewer 1

All modifications are highlighted in yellow in the “foods-1150102 Revised highlighted manuscript”.

Line references in our answers point to the revised highlighted manuscript.

Point 1:
The experimental design for this study was well planned and the manuscript has been well drafted. There are, however, a couple of revisions that the authors need to consider. I have highlighted some comments for the authors below:-

Response 1: First of all, thank you for the positive evaluation.

Point 2:

Introduction. This section has been clearly written and the scientific gap very well-identified.

Response 2: Thank you for the feedback.

Point 3:

L 129, L226, 266, why was Wtech stored at 8 °C and not any other temperature

Response 3: Thank you for this question. All samples were stored at 8 °C (see e.g. lines 156-157). In a previous storage test, the influence of temperature on particle size and viscosity was tested for up to 10 days at 8 °C. No change was shown in the investigated time frame. Also, suspension preparation using the standard method was conducted with overnight storage at 8 °C, whereby results matched those of previous experiments in our lab. Therefore, the same cooling conditions were used for further storage of the samples.

Point 4:

L155, Could the authors please explain how the protein content was adjusted to 5 g 100 g-1 for all three whey concentrates.

Response 4: Thank you for bringing to our attention that this was not clearly written. In lines 150-151, information was added (“The protein rich retentate and the permeate were collected.”) helping to understand the following explanation in lines 154-156, which were also modified to include the following sentence: “The final protein content of the retentate was adjusted to 5 5 (w/w) for all three whey concentrates by diluting it with the permeate from the UF.

Point 5:

L183. How were the samples were diluted to a final protein concentration of 1 % (w/w)? what solvent was used?

Response 5: Thank you for bringing to our attention that the description was not clearly written. The description of the sample preparation for HPLC measurements is now more precisely explained, distinguishing between preparation of whey protein powders and whey protein concentrates (line 183-185). Explanation was improved as follows: “The whey protein powders were dissolved in 99 % (w/w) AqDD to final protein concentrations of 1 % (w/w). The whey protein concentrates with 5 % (w/w) were diluted to 1 % (w/w) by addition of AqDD (w/w).”.

Point 6:

L187. Why -28 °C and not -80 °C or -18 which is the typical temperature for most freezers?

Response 6: Thank you for the question. The method was performed according to Ostertag et al. 2021 (manuscript reference 23), where samples were also stored at -28 °C. The procedure was followed strictly without modifications.

Point 7:

L270. The authors report using a temperature of 18-20 degrees but earlier on they mention 21 degrees. Why were there variations and was there any impact of temperature on the experiments?

Response 7: The particle size was determined by static light scattering using a LS 13 320 laser scattering particle size analyzer (Beckman Coulter LS 13 320 fitted with a Universal Liquid Module and control software v6.01, Beckman Coulter Inc., Miami, FL, USA). The measurements were performed at room temperature reaching from 18 to 20 °C (lines 271 – 273). No influence of temperature on the results is expected and the equipment does not have temperature control. Measurements were conducted in the solvent that was tempered at ambient temperature; hence, the temperature range was provided. The pH was determined at 21 °C since it is commonly referred to as room temperature. The temperature was always measured along with pH, and the pH value was always taken at this temperature. It should be noted that the particle size analysis and pH measurements were conducted in different laboratory rooms.  

Point 8:

L450. The authors mention that the pectin was poorly water-soluble. Could they please quantify this? what was the % solubility of the pectin? And what was the temperature of water used in this case?

Response 8: Thank you for the question thereby bringing to our attention that this sentence was not clearly written. Pectin could be solubilized but needed more time compared to whey protein isolate and whey protein concentrate, thereby limiting hydration time. The sentence was revised as follows: “Furthermore, solid dispersion of whey powder with pectin requiring longer hydration times was associated with the formation of larger complexes (methods C and D).” (line 454-456).

Point 9:

L451, Please consider revising the sentence “This indicating that blending of powders does not improve the solubility to the required extant” to make it clearer.

Response 9: Thank you for the suggestion. The explanation was set up differently to make it clearer (line 456-459), giving: “Often solubility can be improved by blending of two different powders increasing surface area of the poorer soluble compound [30]. However, results indicate that blending of powders does not shorten hydration time of pectin to 24 h.”

Point 10:

L491 “Therefore, use in beverages like drinking yogurt is a possible suitable” Please revise.

Response 10: Thank you for pointing out this mistake. The sentence was revised as follows: “Therefore, use in beverages such as drinkable yogurt is a more suitable application [7].” (line 498-499).

Point 11:

L568 Why do the authors propose investigating phase separation at pH ≤ 3.5 (-)?

Response 11: Thank you for this question. Before application in the technical scale, yield and purity need to be adjusted in the best way possible, to provide a suitable starting point. Therefore, low pH should be considered, possibly resulting in even better yield and purity, due to improved complex aggregation and separation at pH < 3.5. For the decanter in the technical scale, higher throughputs with sample volumes of at least 40 L are necessary to run the experiments, compared to 50 mL in lab scale. Therefore, optimization should be performed in lab scale to limit required sample amounts. An additional introduction sentence (line 578-579: “Before application in the technical scale, yield and purity need to be adjusted in the best way possible, to provide a suitable starting point for upscaling at the decanter.”) as well as a more detailed explanation were added (line 582-583: “possibly resulting in larger aggregates and sharper separation.”).

Reviewer 2 Report

The manuscript entitled "Assessing whey protein sources, dispersion preparation method and enrichment of thermomechanically stabilized whey protein pectin complex for technical scale production” is well written and structured. It is an interesting topic that reporting the use of sustainable products in terms of new applications of whey protein, and new processing methodologies.

General comments and suggestions.

Very extensive title

Line 54 – double paragraph

Line 64 – the abbreviation was previously defined

Line 99 – Materials (only)

Line 145 – when you define an abbreviation, you should always use it

For example “UF”, you should apply it in line 145, 148, 150, 156 (review)

Line 218 – the abbreviation “SSHE” was previously defined

Line 269 – what is the PIDS?!

Line 379 – format spaces in table 4

Studies on particle stability would be interesting.

Author Response

Detailed response to comments of reviewer 2

All modifications are highlighted in yellow in the “foods-1150102 Revised highlighted manuscript”.

Line references in our answers point to the revised highlighted manuscript.

Point 1:

The manuscript entitled "Assessing whey protein sources, dispersion preparation method and enrichment of thermomechanically stabilized whey protein pectin complex for technical scale production” is well written and structured. It is an interesting topic that reporting the use of sustainable products in terms of new applications of whey protein, and new processing methodologies.

Response 1: First of all, thank you for the positive evaluation. The manuscript was proofread and the language was improved, as requested (e.g. lines 58-59, 68-69). We apologize that we could not meet the expectations for background information and references in the introduction. However, since the other reviewer deemed the content as sufficient, we have chosen to not make significant changes to the contents of the introduction. If the reviewer has some specific areas for improvement and references in mind, we would gladly incorporate them into our manuscript.    

Point 2:

Very extensive title

Response 2: We appreciate your comment. Due to the comprehensive process of whey protein pectin complex formation and the different steps of the process that were investigated, the manuscript has been summarized in the title by pointing out the main topics. In all our attempts to shorten the title, essential information was lost. Therefore, we have chosen to not make any changes.

Point 3:

Line 54 – double paragraph

Response 3: Thank you for bringing it to our attention. The double paragraph was removed (line 54).

Point 4:

Line 64 – the abbreviation was previously defined

Response 4: Thank you for bringing it to our attention. The complete designation was removed (line 64).

Point 5:

Line 99 – Materials (only)

Response 5: Thank you for the suggestion. The heading has been changed (line 98).

Point 6:

Line 145 – when you define an abbreviation, you should always use it

For example “UF”, you should apply it in line 145, 148, 150, 156 (review)

Response 6: Thank you for bringing up this point. The abbreviations were added for UF (lines 144, 147, 149, 156), WPP (line 37) and SSHE (lines 24, 561).

Point 7:

Line 218 – the abbreviation “SSHE” was previously defined

Response 7: The abbreviation was deleted (line 219).

Point 8:

Line 269 – what is the PIDS?!

Response 8: Thanks you for bringing up this question. The abbreviation PIDS stands for Polarization Intensity Differential Scattering (PIDS) technology. It is a Beckman Coulter-patented technique that enables direct detection of particles as small as 10 nm, when combined with laser diffraction. The complete designation was added to the manuscript (line 271-272).

Point 9:

Line 379 – format spaces in table 4

Response 9: Thank you for bringing it to our attention. The format of the space was adapted (line 382).

Point 10:

Studies on particle stability would be interesting.

Response 10: We investigated the stability of the WPPC in previous studies by Protte et al. 2017, 2018 (see manuscript reference 9nd 10).